# Outdoor Education, Integrated Soccer Activities, and Learning in Children with Autism Spectrum Disorder: A Project Aimed at Achieving the Sustainable Development Goals of the 2030 Agenda

**Stefania Morsanuto [1,\*], Francesco Peluso Cassese [2], Francesco Tafuri [2] and Domenico Tafuri [3,\*]**

1   Department of Humanistic Sciences, Pegaso University, 80143 Naples, Italy
2   Department of Psychology and Science of Education, University of Study Niccolò Cusano, 00166 Rome, Italy; francesco.peluso@unicusano.it (F.P.C.); francesco.tafuri@unicusano.it (F.T.)
3   Department of Motor and Well-Being Sciences, University of Study Parthenope, 80133 Naples, Italy
\*   Correspondence: stefania.morsanuto@unipegaso.it (S.M.); domenico.tafuri@uniparthenope.it (D.T.)

**Abstract:** This research aims to promote motor activity in children with autism spectrum disorder through the development of an adapted integrated soccer project played outdoors. The project is carried out in collaboration with the nonprofit "Smile Association" of the city of Frosinone (Lazio-Italy) and the Pegaso and Parthenope Universities. The main purpose of the Smile Association is to provide an educational and sports service in an area poorly provided with opportunities for children with intellectual disabilities, allowing them to change their predominantly sedentary condition and, through corporeality, to improve cognitive, affective, and relational processes. Participation in sports activities allows for the development of motor skills, social interactions, and cognitive stimulation. There is a need for available resources and programs dedicated exclusively to children with ASD to help them develop social, motor, and cognitive skills. One of the programs that might be helpful is a structured group play program involving physical activity. In addition, through the project, a dense social network has been activated between the third sector, sports companies, and local health authorities. The project aims to achieve the Sustainable Development Goals of the 2030 Agenda in Articles 3, 4, and 11. This paper describes the correlation between learning and motor activity. According to the hypotheses of this research, integrated soccer activity implements cognitive skills, particularly those related to memory for visual stimuli and to the theory of mind, as well as sense-motor skills. Work carried out to improve attentional skills can reduce stereotypical behaviours. The results showed that continuous and regular engagement in motor activities has positive health outcomes for children with autism in terms of reduced symptomatology and improved quality of life. A total of 108 statistical units (the control group consisted of 18 statistical units, and the sample comprised 90 participants) aged 8 to 11 years were examined. The project is aimed at girls (in compliance with Art. 5 regarding gender equality); however, the participants identified by the local health authority are predominantly male.

**Keywords:** autism; adapted integrated soccer; outdoor; cognitive enhancement; 2030 Agenda

## 1. Introduction

This research aims to promote motor activity in children with autism spectrum disorder through the development of an adapted integrated soccer project [1] carried out outdoors.

The positive influences of sports at the biopsychosocial level in boys with autism spectrum disorder, have been widely highlighted in the literature [2], analysing the impact that motor activities have in boys with autism.

Autism spectrum disorder is a neurodevelopmental disorder characterized by difficulties in social interaction and communication, repetitive behaviours, and narrow interests

present from early childhood. It is a heterogeneous disorder because each child has individual traits and characteristics. According to reports from epidemiological studies, this condition is diagnosed more frequently in boys than girls, with a ratio of 4 to 1 [3]. Different possible hypotheses were formulated to explain the predominance of the male gender in autism diagnoses, each of which was related to different factors. In particular, according to the "female protective effect" (FPE) hypothesis, women would be less vulnerable to autism-related risk factors due to innate protective mechanisms. Considering the same level of severity, it would appear that females with autism have more genetic mutations than males related to large deletions or duplications of DNA tracts. In addition, low levels of testosterone present before and at the time of birth could also play a protective role in the female gender [4].

Education through play, movement, sports, and group activities offers the child with autism a concrete opportunity to acquire early on, in integrated settings, the fundamental assumptions of primary and secondary intersubjectivity, the primary categories of space and time, basic social rules, and appropriate behaviours in different contexts [5]. The variables in the studies examined are related to measures of stereotypic behaviours, attentional behaviours, and finally measures related to social-emotional behaviour [6]. The results showed that continuous and regular engagement in motor activities has positive health outcomes for boys with autism in terms of reduced symptomatology and improved quality of life. In conclusion, motor interventions have been shown to improve several areas: behavioural, emotional, social, and cognitive [7]. Based on a Tel Aviv University study [8], which determines the benefits of outdoor motor activities for individuals with autism, the project's outdoor activities have been adapted.

The research carried out shows that interventions based on outdoor challenges can be effective in reducing the overall severity of autism spectrum disorder (ASD) symptoms. Significant improvements in social cognition, social motivation, and stereotypical behaviours were found in the sample examined [9]. It is necessary, therefore, to have resources and programs dedicated exclusively to children with ASD to help them develop social, motor, and cognitive skills [7,10].

The project presented not only supports the literature research, but also supports its constructs with an educational intervention for a large group of children. Compared to the studies reviewed, the present one has the characteristic of continuity. The data collected for the present research are still evolving and may provide important findings in the future. The results have shown that continuous and regular engagement in motor activities has positive health outcomes for children with autism in terms of reduced symptoms and improved quality of life. The intervention was made effective by a network developed between the local health authority that identified the children, the third sector that implemented the educational proposal, the territorial services that made the sports facilities available, and the universities that supervised the pedagogical and scientific activities in line with public engagement

The project presented responds to some specific goals of the 2030 Agenda. An action agenda for sustainable development was signed in 2015 by 193 member countries of the UN, in which 17 Sustainable Development Goals—SDG to be achieved by the year 2030 were identified. In particular, we refer to Goal No. 4, which aims to ensure inclusive and equitable quality education and promote lifelong learning opportunities for all. The international community recalls the importance of quality education and training to improve the living conditions of individuals, communities, and societies. Target 11.3 in the sustainable development goals sets out to enhance inclusive urbanization. In the Agenda, moreover, Goal n. 3 refers to the themes of health and well-being specifically, "Any form of physical activity which, through organized or unorganized participation, has as its objective the improvement of physical and mental conditions, the development of social relations, or the consequence of achievements in the course of competitions at all levels". The project is aimed at girls (in compliance with Art. 5 regarding gender equality); however, the participants identified by the local health authority are predominantly male.

## 2. Autism and Motor Activity

The project adheres to the concept of empowerment in sports for people with disabilities [11,12], resting on the awareness of one's own skills and perception of self-efficacy. The goal is to achieve, through the sports experience, better control over personal resources and the environment in which one lives [6]. Hutzler's model postulates that sport activity results in a number of psychological and social benefits in people with disabilities, starting with the assumptions: motor performance drives functional efficiency; successful experiences improve self-efficacy; improved body confidence improves physical self-concept and self-esteem; mood and affective disorders become lighter; and growth in skill level leads to better social acceptance. The activity adheres to the directions of adapted physical activity [12], according to which an inclusive sports project: must provide a service; must have an academic specialization with interdisciplinary fields of knowledge; needs a philosophy or set of beliefs, which guide practical activities, and an attitude of acceptance, which predisposes to various behaviors; must be based on a dynamic system of interacting theories and practices; must have a process and a product; and must have a network dedicated to disability rights [13].

Peer-mediated strategies can be especially helpful to train recreational skills; such procedures allow employing target skills with peers [14,15].

Existing data suggest that behavioral skills training (BST) and peer-mediated intervention are effective in promoting targeted motor and social skills, both when implemented individually and in tandem [16]. Behavioral skills training (BST) is a method used to train educators. BST is a combination of performance and competency with a particular skill or set of skills and provides clear and concise instructions, which means that the instructions are easy to remember and highlight key teaching points. Moreover, it provides visual instructions as a reminder [15].

## 3. Autism and Outdoor Education

The study led by Ditza Antebi-Zachor demonstrates the positive impact of outdoor physical activity in children with autism. Children from seven special education institutions in Tel Aviv took part in the research. They underwent weekly outdoor physical activity sessions with specialized instructors. The study shows that outdoor motor activities improve autism symptoms and social communication skills [17]. The study is confirmed by similar Italian research works [18].

A global brain analysis revealed that the time spent outdoors was positively associated with grey matter volume in the right dorsolateral prefrontal cortex [19].

Furthermore, a persistent positive effect was also shown at the conclusion of physical activity associated with fluid intake, leisure time, and hours of sun exposure.

This sudden evidence could be traced back to fractional anisotropic (FA) measurements that assess the quantity and movement of water molecules in white matter: higher values indicate that the movement of molecules is more directional, while lower values indicate that their movement is more diffuse. Autistic children's brains showed a higher mean FA than their typical peers. This is consistent with findings showing that the brains of children with autism grow abnormally fast. These findings could stem from a presumable correlation between increased white matter and increased AF [20]. Gray and white matter are closely related, with the former having the function of selecting and initiating information that travels along the nervous system, but also serving as the starting point of motor inputs. White matter presides over the connection and interaction of motor stimuli. Outdoor activities, therefore, can improve FA performance and enhance brain plasticity [21].

Furthermore, the use of peer-mediated BST for discrete sporting ability in individuals with disabilities is a procedure that may hold particular promise because a discrete skill could be taught and reinforced in a naturalistic context [15].

## 4. Hypothesis and Goals

According to the hypothesis of this research, integrated soccer activity implements cognitive skills, particularly those related to memory for visual stimuli and to the theory of mind, as well as sensory motor skills. Improving attentional skills can reduce stereotypical behaviours (see for illustration the literature [22–25] composite index studies; refs. [26,27]). Difficulties in shifting attention between external stimuli and internal dialogue, according to Anderson, could be the result of poor communication between regions of the attention network [28]. Therefore, in response to the hypothesis formulated, we want to demonstrate the development of children's cognitive-behavioural skills through motor activity; contextually, we want to verify that the same skills are transferred to other areas of everyday life.

## 5. Method

The project was carried out in collaboration with the nonprofit "Smile Association" of the city of Frosinone (Lazio-Italy) and the Pegaso and Parthenope Universities, the Local Health Authority (AsL) of Frosinone, the Municipality of Ceprano, and the Frosinone Soccer Club with the sponsorship of the Lazio Region (Italy). The project is funded by several social promotion calls and has been ongoing since 2020. The main purpose of the project is to promote motor activity in children with autism spectrum disorder through integrated soccer. The associations' main goal is to provide an educational and sports service in an area that is poorly provided with opportunities for children with intellectual disabilities, allowing them to modify their predominantly sedentary condition and, through corporeality, to improve cognitive, affective, and relational processes.

The project involves children from 8 to 11 years old with autism spectrum disorder, diagnosed by professionals at the Local Health Authority of Frosinone, through the fourth version of the Wechsler Intelligent Scale for Children [25], which allows for assessing the cognitive abilities of children and young people. The WISC-IV assesses four cognitive areas through separate cognitive indices: Verbal Comprehension Index, Visuo-Perceptive Reasoning Index, Working Memory Index, and Processing Speed Index. This provides basic benchmarks for the development of educational pathways. There is, in addition, a control group homogeneous to the sample that did not participate in the proposed integrated soccer activity.

A total of 108 statistical units were examined. The control group consists of 18 statistical units, and the sample comprised 90 participants. The project is aimed at girls; however, the participants identified by the local health authority are predominantly male.

Sports activities take place every two weeks for a total of four hours, from two years ago to today (the project is still ongoing). Integrated football is played 5 against 5. The activity is offered free of charge. Integrated football features four goals instead of two, a smaller and/or lighter ball in certain phases of the game, and the use of protected areas for side goals. Each player has a role defined by his or her motor and cognitive skills and consequently has a direct opponent of the same level. There are five roles defined by athletic compatibility rules. Within the team, there is the role of the mentor, who coordinates mutual aid among the athletes.

Among the aspects that make the activity useful and relevant for children with autism spectrum disorders is the use of precise routines, such as always carrying out the activity on the same days and at the same times during the week or getting dressed and undressed within a recognizable and well-defined space such as the locker room. The structure of the interventions foresees the development of the personal autonomy of the children in the changing rooms. All personal materials, spaces, and procedures (e.g., dressing, preparation of the water bottle, preparation of the material) are marked with images. The activities include a short initial briefing aimed at the memory recovery of the previous lesson, followed by the warm-up and the match.

The example model is the BST, which is made up of four parts: instruction, modelling, testing, and feedback. (1) The instructions given and the description of the skill should be

clear, concise and as pragmatic as possible. It is also important to give them a rationale for the skill and the reasons why you are teaching it. Communication is clear and simple. Verbal language is not used exclusively, but the student is also provided with visual explanations that become models available to the child throughout the activity [29,30]. (2) Modelling shows how to perform the skill. (3) During the test, the child is allowed to practice the new skills. Comparison with peers is important at this stage. If necessary, they go back to modelling to make sure they have the necessary skills. (4) Feedback can be hard to give and hard to get, so positive reinforcement is of paramount importance [7].

On-field activities are managed by five professional educators and five specialist instructors in preventive and adapted motor activities, coordinated by a pedagogist. University professors are responsible for monitoring the educational proposal and coordinating the research. Validated tests were used [31]. The tests were also administered to a control group, homogeneous in terms of age and disorder, that did not participate in the project. This guarantees compliance with the legislation on the protection of personal data (and informs that the personal data provided by the interested parties through the direct collection channel pursuant to articles 13, paragraph 1, and 14, paragraph 1, of the Regulation European No. 679/2016—General Data Protection Regulation, from now on GDPR).

The analysis of the findings of the experimental design does not only consist of a data processing and data synthesis phase but necessarily requires prior reflection on compliance with the requirements of internal and external validity, without which the entire subsequent course of action would at the very least be lacking if not downright useless in the event of a violation of the basic assumptions. The assertion of the internal validity of the experiment is to be understood as the possibility of defending the imputation of causality towards experimental variable X, constituted in our case by the training intervention on the subject in question, from the influence of other external factors, the influence of which, if not controlled, is capable of producing experimental results that cannot be interpreted according to the intentions of the researcher, or of 'invalidating', in fact, the experiment tout court. Once the robustness of the experiment has been ascertained from an internal point of view, it is then possible to proceed to the generalization of the results of the experiment or, if one prefers, to assess its degree of representativeness with respect to all those subjects who were not involved in the experiment and to whom we would like to extend our focus. This type of judgement is referred to as external validity [32].

### 6. Tools

Participants are monitored through to the NEPSY II assessment battery [2], which is administered at the beginning and end of the educational year (T0 and T1). NEPSY-II provides a neuropsychological assessment of the cognitive abilities of subjects from 3 to 16 years of age in relation to specific cognitive domains. The tool allows both for a global assessment and for developing a survey aimed at one or more domains and is able to ascertain the cognitive abilities or typical disorders generally diagnosable for the first time during childhood. The NEPSY-II, therefore, allows for making an accurate diagnosis and planning the interventions necessary for recovery and for reaching full functionality both at school and at home. The battery is validated [31,33] (also in Italy) and consists of 33 tests that refer to six different cognitive domains: Attention and executive functions, language, memory and learning, sensory-motor functions, social perception, and visuo-spatial processing. For the purposes of this research, the following sub-tests were administered:

1. M3, which assesses learning and visuospatial memory. A deferred test assesses long-term visuospatial memory and visual discrimination ability. Memory for visual stimuli and memory for spatial location are assessed separately.
2. M6, which assesses narrative memory based on both free or guided reenactment and recognition conditions.
3. SM1, which assesses the child's speed and ability to perform repetitive finger movements and to execute quickly in motor program for sequences of movements.

4.  S01, based on the theory of mind and consisting of two parts: the first verbal one assessing the ability to understand mental constructs such as beliefs, intentions, deceptions, fantasy, and pretense, as well as the ability to recognize and contextualize emotions, and the fact that others may have different thoughts, ideas, and feelings from ours. The second nonverbal type assesses the ability to understand how emotions connect to a social context and to recognize the appropriate state of mind. The study did not respond to requests for approval from the ethics committee that received notice of the research. All tests were authorized by parents who could opt out of the research at any time. The tests are not for diagnostic purposes.

## 7. Logistical Resources

In line with Article 11 of the 2030 Agenda, the spaces provided were immediately made "inclusive, safe and sustainable". All activities take place outdoors.

Several outdoor sports facilities have been used for the development of the project: An A11 soccer field, a multipurpose sports field for training, and an athletic track. All necessary sports equipment and recognition materials are provided for the children's personal use: uniforms, bibs, and visual aids for mnestic enhancement.

## 8. Results

The data analysis was carried out with IBM's SPSS statistical analysis software. The control group consists of 18 statistical units, and the sample comprises 90 participants. There are a total of 108 statistical units (Table 1). The project is aimed at girls (in compliance with Art. 5 regarding gender equality); however, the participants identified by the local health authority are predominantly male.

**Table 1.** Sample and control groups frequency.

|  |  | Frequency | Percent | Valid Percent | Cumulative Percent |
|---|---|---|---|---|---|
| Valid | Control Group | 18 | 16.7 | 16.7 | 16.7 |
|  | Sample Group | 90 | 83.3 | 83.3 | 100.0 |
|  | Total | 108 | 100.0 | 100.0 |  |

The independent samples *t*-test compares the means of two independent groups in order to determine whether there is statistical evidence that the associated population means are significantly different. The independent samples *t*-test is a parametric test.

By comparing the sample with the control group through an independent samples *t*-test at time T0, it is found that the two groups have no statistically significant differences in means.

The null hypothesis for the independent samples *t*-test is $H_0$: the population averages of the two groups (control and sample) are equal (i.e., $\mu1 = \mu2$).

The test is robust to the hypothesis that the variances of the two groups are homogeneous. One can use Levene's test for the equality of the variances, yielding two *t*-tests for independent samples calculated differently that will give a valid result regardless of whether this assumption is met or violated practically; the system provides one *t*-test for independent samples that is calculated normally (with pooled variances) and another for when the assumption uses separate variances (i.e., ungrouped variances) and the Welch–Satterthwaite correction to the degrees of freedom is violate. Levene's test indicates equality of variances; thus, by taking the values in the first row, we see that the *p*-value is always greater than 0.05. Therefore, the null hypothesis, which implies that the differences between the means (obtained in the various tests) cannot be rejected, is not statistically significant. The value 0 is then always within the confidence intervals (lower and upper). This is another indicator that the null hypothesis cannot be rejected and that, therefore, there are no statistically significant differences between the means of the results of the two groups (Table 2).

**Table 2.** Independent samples *t*-test at T0—control vs. sample.

| | | Levene's Test for Equality of Variances | | *t*-Test for Equality of Means | | | | | 95% Confidence Interval of the Difference | |
| --- | --- | --- | --- | --- | --- | --- | --- | --- | --- | --- |
| | | F | Sig. | t | df | Sig. (2-Tailed) | Mean Difference | Std. Error Difference | Lower | Upper |
| T0_M3—Total Immediate | Equal variances assumed | 0.012 | 0.913 | −0.941 | 106 | 0.349 | −0.956 | 1.016 | −2.969 | 1.058 |
| | Equal variances not assumed | | | −0.959 | 24.738 | 0.347 | −0.956 | 0.997 | −3.009 | 1.098 |
| T0_M3—Total Deferred | Equal variances assumed | 0.304 | 0.582 | −1.241 | 106 | 0.217 | −1.111 | 0.895 | −2.886 | 0.663 |
| | Equal variances not assumed | | | −1.339 | 26.273 | 0.192 | −1.111 | 0.830 | −2.816 | 0.594 |
| T0_SM1—Finger Tapping | Equal variances assumed | 0.569 | 0.452 | −1.363 | 106 | 0.176 | −1.389 | 1.019 | −3.409 | 0.631 |
| | Equal variances not assumed | | | −1.498 | 26.852 | 0.146 | −1.389 | 0.927 | −3.291 | 0.513 |
| T0_SO1—Theory of Mind | Equal variances assumed | 0.158 | 0.692 | −1.046 | 106 | 0.298 | −1.022 | 0.977 | −2.960 | 0.915 |
| | Equal variances not assumed | | | −1.078 | 25.031 | 0.291 | −1.022 | 0.948 | −2.975 | 0.930 |
| T0_M7—Phrase repetition | Equal variances assumed | 0.002 | 0.961 | −0.796 | 106 | 0.428 | −0.778 | 0.978 | −2.716 | 1.161 |
| | Equal variances not assumed | | | −0.793 | 24.218 | 0.436 | −0.778 | 0.981 | −2.801 | 1.246 |
| T0_M6—Narrative Memory—Spontaneous Recollection | Equal variances assumed | 0.109 | 0.742 | −0.943 | 106 | 0.348 | −0.867 | 0.919 | −2.689 | 0.956 |
| | Equal variances not assumed | | | −0.990 | 25.509 | 0.332 | −0.867 | 0.875 | −2.668 | 0.935 |
| T0_M6—Narrative Memory—Total Recollection | Equal variances assumed | 0.431 | 0.513 | −1.174 | 106 | 0.243 | −0.967 | 0.824 | −2.600 | 0.666 |
| | Equal variances not assumed | | | −1.246 | 25.829 | 0.224 | −0.967 | 0.776 | −2.561 | 0.628 |
| T0_M6—Narrative Memory—Total Recognition | Equal variances assumed | 0.140 | 0.709 | −1.132 | 106 | 0.260 | −1.100 | 0.971 | −3.026 | 0.826 |
| | Equal variances not assumed | | | −1.168 | 25.046 | 0.254 | −1.100 | 0.942 | −3.039 | 0.839 |

Moreover, by comparing the sample group with the control group through an independent samples *t*-test of the control group at time T1, we find that the sample group always obtained higher values than the control group. Choosing the correct row based on the result of Levene's test (green cells) on the equality of variances, it is found that the *p*-value is greater than 0.05 (yellow cells) for the SM1—Finger Tapping and M6—Narrative Memory—Total Recognition tests. Therefore, we can reject the null hypothesis and conclude that for these two tests, the differences between the means of the sample and control groups are statistically significant. For the two tests, the result is corroborated by the fact that the 0 value falls outside the confidence intervals (lower and upper—blue cells). This indicates that the null hypothesis can be rejected, and thus there are statistically significant differences between the mean results of the two groups (Table 3).

**Table 3.** Independent samples *t*-test at T1—control vs. sample.

| | | Levene's Test for Equality of Variances | | *t*-Test for Equality of Means | | | | | 95% Confidence Interval of the Difference | |
| --- | --- | --- | --- | --- | --- | --- | --- | --- | --- | --- |
| | | F | Sig. | t | df | Sig. (2-Tailed) | Mean Difference | Std. Error Difference | Lower | Upper |
| T1_M3—Total Immediate | Equal variances assumed | 0.088 | 0.768 | −1.382 | 106 | 0.170 | −1.433 | 1.037 | −3.489 | 0.622 |
| | Equal variances not assumed | | | −1.401 | 24.615 | 0.174 | −1.433 | 1.023 | −3.542 | 0.675 |

**Table 3.** *Cont.*

| | | Levene's Test for Equality of Variances | | *t*-Test for Equality of Means | | | | | | 95% Confidence Interval of the Difference | |
| --- | --- | --- | --- | --- | --- | --- | --- | --- | --- | --- | --- |
| | | **F** | **Sig.** | **t** | **df** | **Sig. (2-Tailed)** | **Mean Difference** | **Std. Error Difference** | | **Lower** | **Upper** |
| T1_M3—Total Deferred | Equal variances assumed | 1.194 | 0.277 | −1.839 | 106 | 0.069 | −1.822 | 0.991 | | −3.786 | 0.142 |
| | Equal variances not assumed | | | −2.147 | 28.995 | 0.040 | −1.822 | 0.849 | | −3.558 | −0.087 |
| T1_SM1—Finger Tapping | Equal variances assumed | 13.957 | 0.000 | −2.867 | 106 | 0.005 | −2.867 | 1.000 | | −4.849 | −0.884 |
| | Equal variances not assumed | | | −3.952 | 38.362 | 0.000 | −2.867 | 0.725 | | −4.335 | −1.399 |
| T1_SO1—Theory of Mind | Equal variances assumed | 0.681 | 0.411 | −1.645 | 106 | 0.103 | −1.600 | 0.973 | | −3.528 | 0.328 |
| | Equal variances not assumed | | | −1.903 | 28.642 | 0.067 | −1.600 | 0.841 | | −3.320 | 0.120 |
| T1_M7—Phrase repetition | Equal variances assumed | 1.778 | 0.185 | −1.520 | 106 | 0.132 | −1.500 | 0.987 | | −3.457 | 0.457 |
| | Equal variances not assumed | | | −1.657 | 26.607 | 0.109 | −1.500 | 0.905 | | −3.358 | 0.358 |
| T1_M6—Narrative Memory—Spontaneous Recollection | Equal variances assumed | 0.094 | 0.760 | −1.311 | 106 | 0.193 | −1.389 | 1.059 | | −3.489 | 0.711 |
| | Equal variances not assumed | | | −1.345 | 24.901 | 0.191 | −1.389 | 1.033 | | −3.516 | 0.739 |
| T1_M6—Narrative Memory—Total Recollection | Equal variances assumed | 0.010 | 0.922 | −1.588 | 106 | 0.115 | −1.456 | 0.917 | | −3.273 | 0.362 |
| | Equal variances not assumed | | | −1.593 | 24.368 | 0.124 | −1.456 | 0.914 | | −3.340 | 0.429 |
| T1_M6—Narrative Memory—Total Recognition | Equal variances assumed | 1.332 | 0.251 | −2.019 | 106 | 0.046 | −1.956 | 0.969 | | −3.876 | −0.035 |
| | Equal variances not assumed | | | −2.322 | 28.416 | 0.028 | −1.956 | 0.842 | | −3.680 | −0.231 |

Comparing the control group with the sample through a paired samples *t*-test at times T0 and T1, we find that there are no statistically significant differences between the two time points. Thus, the behavior of the control group remains the same at the two times (Table 4).

**Table 4.** Paired samples *t*-test for the control group—T0 vs. T1.

| | | Paired Differences | | | | | | | |
| --- | --- | --- | --- | --- | --- | --- | --- | --- | --- |
| | | | | | 95% Confidence Interval of the Difference | | | | |
| | | **Mean** | **Std. Deviation** | **Std. Error Mean** | **Lower** | **Upper** | **t** | **df** | **Sig. (2-Tailed)** |
| Pair 1 | T0-T1 M3—Total Immediate | 0.000 | 2.196 | 0.518 | −1.092 | 1.092 | 0.000 | 17 | 1.000 |
| Pair 2 | T0-T1 M3—Total Deferred | −0.278 | 1.841 | 0.434 | −1.193 | 0.638 | −0.640 | 17 | 0.531 |
| Pair 3 | T0-T1 SM1—Finger Tapping | −0.278 | 2.445 | 0.576 | −1.494 | 0.938 | −0.482 | 17 | 0.636 |
| Pair 4 | T0-T1 SO1—Theory of Mind | −0.056 | 1.162 | 0.274 | −0.633 | 0.522 | −0.203 | 17 | 0.842 |
| Pair 5 | T0-T1 M7—Phrase repetition | 0.667 | 2.376 | 0.560 | −0.515 | 1.848 | 1.190 | 17 | 0.250 |
| Pair 6 | T0-T1 M6—Narrative Memory—Spontaneous Recollection | 0.333 | 2.497 | 0.589 | −0.908 | 1.575 | 0.566 | 17 | 0.579 |
| Pair 7 | T0-T1 M6—Narrative Memory—Total Recollection | 0.000 | 2.114 | 0.498 | −1.051 | 1.051 | 0.000 | 17 | 1.000 |
| Pair 8 | T0-T1 M6—Narrative Memory—Total Recognition | −0.278 | 0.958 | 0.226 | −0.754 | 0.199 | −1.230 | 17 | 0.236 |

Comparing the sample group with the control group through a paired samples *t*-test of the sample group at times T0 and T1, we find that for all tests the averages improved. In particular, as the *p*-values are smaller than 0.05, we can say that these differences are statistically significant for the tests (Table 5):

- M3—Total Deferred;
- SM1—Finger Tapping;
- SO1—Theory of Mind;
- M6—Narrative Memory—Total Recognition.

**Table 5.** Paired samples *t*-test for the sample group—T0 vs. T1.

| | | **Paired Differences** | | | | | | | |
| --- | --- | --- | --- | --- | --- | --- | --- | --- | --- |
| | | | | | **95% Confidence Interval of the Difference** | | | | |
| | | **Mean** | **Std. Deviation** | **Std. Error Mean** | **Lower** | **Upper** | ***t*** | **df** | **Sig. (2-Tailed)** |
| Pair 1 | T0-T1 M3—Total Immediate | −0.478 | 3.282 | 0.346 | −1.165 | 0.210 | −1.381 | 89 | 0.171 |
| Pair 2 | T0-T1 M3—Total Deferred | −0.989 | 3.337 | 0.352 | −1.688 | −0.290 | −2.811 | 89 | 0.006 |
| Pair 3 | T0-T1 SM1—Finger Tapping | −1.756 | 4.611 | 0.486 | −2.721 | −0.790 | −3.612 | 89 | 0.001 |
| Pair 4 | T0-T1 SO1—Theory of Mind | −0.633 | 2.985 | 0.315 | −1.258 | −0.008 | −2.013 | 89 | 0.047 |
| Pair 5 | T0-T1 M7—Phrase repetition | −0.056 | 3.552 | 0.374 | −0.799 | 0.688 | −0.148 | 89 | 0.882 |
| Pair 6 | T0-T1 M6—Narrative Memory—Spontaneous Recollection | −0.189 | 3.277 | 0.345 | −0.875 | 0.497 | −0.547 | 89 | 0.586 |
| Pair 7 | T0-T1 M6—Narrative Memory—Total Recollection | −0.489 | 2.833 | 0.299 | −1.082 | 0.105 | −1.637 | 89 | 0.105 |
| Pair 8 | T0-T1 M6—Narrative Memory—Total Recognition | −1.133 | 3.471 | 0.366 | −1.860 | −0.406 | −3.097 | 89 | 0.003 |

Studying the correlations between the T0 and T1 times, we see that there are strong positive correlations between the M6—Narrative Memory—Spontaneous Recollection and M6—Narrative Memory—Total Recollection tests toward all other tests and that this correlation strengthens over time. As the mean values of these two tests improve, it is possible to expect improvements in the other tests as well. In contrast, the SO1—Theory of Mind test loses (or weakens) its positive correlation with the M3 and SM1 tests. The SM1—Finger Tapping test, on the other hand, goes from having no correlation with the M3 test to having a weakly positive one (Tables 6 and 7).

**Table 6.** Pearson's product–moment correlation—T0—Sample group.

| | | **T0_M3—Total Immediate** | **T0_M3—Total Deferred** | **T0_SM1—Finger Tapping** | **T0_SO1—Theory of Mind** | **T0_M7—Phrase Repetition** | **T0_M6—Narrative Memory—Spontaneous Recollection** | **T0_M6—Narrative Memory—Total Recollection** | **T0_M6—Narrative Memory—Total Recognition** |
| --- | --- | --- | --- | --- | --- | --- | --- | --- | --- |
| T0_M3—Total Immediate | Pearson Correlation | 1 | 0.853 (**) | −0.075 | 0.170 | −0.007 | 0.328 (**) | 0.251 (*) | 0.184 |
| | Sig. (2-tailed) | | 0.000 | 0.481 | 0.110 | 0.951 | 0.002 | 0.017 | 0.083 |
| | N | 90 | 90 | 90 | 90 | 90 | 90 | 90 | 90 |
| T0_M3—Total Deferred | Pearson Correlation | 0.853 (**) | 1 | −0.020 | 0.244 (*) | 0.074 | 0.332 (**) | 0.261 (*) | 0.223 (*) |
| | Sig. (2-tailed) | 0.000 | | 0.850 | 0.020 | 0.491 | 0.001 | 0.013 | 0.034 |
| | N | 90 | 90 | 90 | 90 | 90 | 90 | 90 | 90 |
| T0_SM1—Finger Tapping | Pearson Correlation | −0.075 | −0.020 | 1 | 0.446 (**) | 0.586 (**) | 0.477 (**) | 0.560 (**) | 0.514 (**) |
| | Sig. (2-tailed) | 0.481 | 0.850 | | 0.000 | 0.000 | 0.000 | 0.000 | 0.000 |
| | N | 90 | 90 | 90 | 90 | 90 | 90 | 90 | 90 |
| T0_SO1—Theory of Mind | Pearson Correlation | 0.170 | 0.244 (*) | 0.446 (**) | 1 | 0.627 (**) | 0.471 (**) | 0.524 (**) | 0.420 (**) |
| | Sig. (2-tailed) | 0.110 | 0.020 | 0.000 | | 0.000 | 0.000 | 0.000 | 0.000 |
| | N | 90 | 90 | 90 | 90 | 90 | 90 | 90 | 90 |
| T0_M7—Phrase repetition | Pearson Correlation | −0.007 | 0.074 | 0.586 (**) | 0.627 (**) | 1 | 0.573 (**) | 0.593 (**) | 0.602 (**) |
| | Sig. (2-tailed) | 0.951 | 0.491 | 0.000 | 0.000 | | 0.000 | 0.000 | 0.000 |
| | N | 90 | 90 | 90 | 90 | 90 | 90 | 90 | 90 |
| T0_M6—Narrative Memory—Spontaneous Recollection | Pearson Correlation | 0.328 (**) | 0.332 (**) | 0.477 (**) | 0.471 (**) | 0.573 (**) | 1 | 0.902 (**) | 0.525 (**) |
| | Sig. (2-tailed) | 0.002 | 0.001 | 0.000 | 0.000 | 0.000 | | 0.000 | 0.000 |
| | N | 90 | 90 | 90 | 90 | 90 | 90 | 90 | 90 |
| T0_M6—Narrative Memory—Total Recollection | Pearson Correlation | 0.251 (*) | 0.261 (*) | 0.560 (**) | 0.524 (**) | 0.593 (**) | 0.902 (**) | 1 | 0.607 (**) |
| | Sig. (2-tailed) | 0.017 | 0.013 | 0.000 | 0.000 | 0.000 | 0.000 | | 0.000 |
| | N | 90 | 90 | 90 | 90 | 90 | 90 | 90 | 90 |
| T0_M6—Narrative Memory—Total Recognition | Pearson Correlation | 0.184 | 0.223 (*) | 0.514 (**) | 0.420 (**) | 0.602 (**) | 0.525 (**) | 0.607 (**) | 1 |
| | Sig. (2-tailed) | 0.083 | 0.034 | 0.000 | 0.000 | 0.000 | 0.000 | 0.000 | |
| | N | 90 | 90 | 90 | 90 | 90 | 90 | 90 | 90 |

** Correlation is significant at the 0.01 level (2-tailed). * Correlation is significant at the 0.05 level (2-tailed).

**Table 7.** Pearson's product–moment correlation —T1—Sample group.

| | | T0_M3— Total Immediate | T0_M3— Total Deferred | T0_SM1— Finger Tapping | T0_SO1— Theory of Mind | T0_M7— Phrase Repetition | T0_M6— Narrative Memory— Spontaneous Recollection | T0_M6— Narrative Memory— Total Recollection | T0_M6— Narrative Memory— Total Recognition |
|---|---|---|---|---|---|---|---|---|---|
| T0_M3—Total Immediate | Pearson Correlation | 1 | 0.893 (**) | 0.221 (*) | 0.113 | −0.008 | 0.390 (**) | 0.380 (**) | 0.099 |
| | Sig. (2-tailed) | | 0.000 | 0.036 | 0.288 | 0.942 | 0.000 | 0.000 | 0.355 |
| | N | 90 | 90 | 90 | 90 | 90 | 90 | 90 | 90 |
| T0_M3—Total Deferred | Pearson Correlation | 0.893 (**) | 1 | 0.224 (*) | 0.154 | 0.052 | 0.337 (**) | 0.323 (**) | 0.144 |
| | Sig. (2-tailed) | 0.000 | | 0.034 | 0.147 | 0.628 | 0.001 | 0.002 | 0.176 |
| | N | 90 | 90 | 90 | 90 | 90 | 90 | 90 | 90 |
| T0_SM1—Finger Tapping | Pearson Correlation | 0.221 (*) | 0.224 (*) | 1 | 0.216 (*) | 0.483 (**) | 0.581 (**) | 0.517 (**) | 0.389 (**) |
| | Sig. (2-tailed) | 0.036 | 0.034 | | 0.041 | 0.000 | 0.000 | 0.000 | 0.000 |
| | N | 90 | 90 | 90 | 90 | 90 | 90 | 90 | 90 |
| T0_SO1—Theory of Mind | Pearson Correlation | 0.113 | 0.154 | 0.216 (*) | 1 | 0.630 (**) | 0.408 (**) | 0.388 (**) | 0.317 (**) |
| | Sig. (2-tailed) | 0.288 | 0.147 | 0.041 | | 0.000 | 0.000 | 0.000 | 0.002 |
| | N | 90 | 90 | 90 | 90 | 90 | 90 | 90 | 90 |
| T0_M7—Phrase repetition | Pearson Correlation | −0.008 | 0.052 | 0.483 (**) | 0.630 (**) | 1 | 0.695 (**) | 0.664 (**) | 0.531 (**) |
| | Sig. (2-tailed) | 0.942 | 0.628 | 0.000 | 0.000 | | 0.000 | 0.000 | 0.000 |
| | N | 90 | 90 | 90 | 90 | 90 | 90 | 90 | 90 |
| T0_M6—Narrative Memory—Spontaneous Recollection | Pearson Correlation | 0.390 (**) | 0.337 (**) | 0.581 (**) | 0.408 (**) | 0.695 (**) | 1 | 0.940 (**) | 0.468 (**) |
| | Sig. (2-tailed) | 0.000 | 0.001 | 0.000 | 0.000 | 0.000 | | 0.000 | 0.000 |
| | N | 90 | 90 | 90 | 90 | 90 | 90 | 90 | 90 |
| T0_M6—Narrative Memory—Total Recollection | Pearson Correlation | 0.380 (**) | 0.323 (**) | 0.517 (**) | 0.388 (**) | 0.664 (**) | 0.940 (**) | 1 | 0.523 (**) |
| | Sig. (2-tailed) | 0.000 | 0.002 | 0.000 | 0.000 | 0.000 | 0.000 | | 0.000 |
| | N | 90 | 90 | 90 | 90 | 90 | 90 | 90 | 90 |
| T0_M6—Narrative Memory—Total Recognition | Pearson Correlation | 0.099 | 0.144 | 0.389 (**) | 0.317 (**) | 0.531 (**) | 0.468 (**) | 0.523 (**) | 1 |
| | Sig. (2-tailed) | 0.355 | 0.176 | 0.000 | 0.002 | 0.000 | 0.000 | 0.000 | |
| | N | 90 | 90 | 90 | 90 | 90 | 90 | 90 | 90 |

** Correlation is significant at the 0.01 level (2-tailed). * Correlation is significant at the 0.05 level (2-tailed).

## 9. Discussion

Through this work, we sought to investigate whether football activity can integrate and implement cognitive skills, in particular those related to memory for visual stimuli and theory of mind, as well as sense-motor skills. Thus, an attempt was made to demonstrate the development of a children's cognitive-behavioral skills through motor activity; at the same time, the aim was to verify that the same skills are transferred to other areas of everyday life. Sub-tests of the NEPSY II battery (used in the diagnostic field) were used to support the research. NESPY II is a nationally and internationally validated instrument. The sub-tests used to investigate learning and visuospatial memory, narrative memory, a child's speed and ability to perform repetitive finger movements and to execute quickly in motor program for sequences of movements, and the theory of mind.

The sample was chosen through collaboration with the local health authority, which is in charge of both the diagnosis and care of children with autism spectrum disorder. To reduce the risk of bias (systematic errors) during the planning, execution, and analysis of data, (internal validity) a sampling plan with precise parameters has been agreed upon. The selection parameters requested from the health authority are homogeneous age groups, the same diagnostic tool, and similar diagnostic values. All children are between 8 and 10 years old and diagnosed through the WISC-IV diagnostic tool. The WISC-IV assesses four cognitive areas by means of separate cognitive indices: Verbal Comprehension Index (VCI), Visual-Perceptual Reasoning Index (IRP), Working Memory Index (IML), and Processing Speed Index (IVE). All selected children have similar (starting) values in the four parameters. The aim of the survey is to know and better understand a certain aspect of autism spectrum disorder. This was achieved by analyzing the data collected so far. The project, however, is still active in the area, as is the data collection, in order to enlarge the sample and make the detection of learning developments more precise.

When comparing the sample and the control group at the time T1, it is found that the sample always obtained higher values than the control group. In particular, it is found that the *p*-value is greater than 0.05 for the SM1—Finger Tapping and M6—Narrative Memory—Total Recognition tests. For these two tests, the differences between the sample and control group averages are statistically significant. By comparing the control group with the sample group through a paired samples *t*-test of the control group at times T0 and

T1, it is found that there are no statistically significant differences between the two times. While the behaviour of the control group remains unchanged at the two times, we find that the averages of the sample group at times T0 and T1, for all tests, improved. These differences are statistically significant for the tests.

Studying the correlations between the T0 and T1 periods, we see that there are strong positive correlations between the M6—Narrative Memory—Spontaneous Recollection and M6—Narrative Memory—Total Recollection tests toward all other tests and that this correlation strengthens over time. When the mean values of these two tests improve, it is possible to expect improvements in the other tests as well. In contrast, the SO1—TOM test weakens its positive correlation with the M3 and SM1 tests. The SM1—Finger Tapping test, instead changes from having no correlation with the M3 tests to having a weakly positive one. The data analysed confirm that the starting hypothesis is achievable through a structured intervention over time.

Both tests related to memory and the motor accuracy of task execution have a positive and statistically significant cut off. It can be hypothesized that, based on the characteristics of children's ASD diagnoses, if attentional skills shift to situation mentalization, mnestic skills are impaired.

The critical aspects of the project include the need to select more 'threats' to internal validity such as confounding variables (random variables influencing both dependent and independent variables, causing a spurious association) or possible experimenter or subject error due to personal situations (e.g., spectrum disorder characteristics are greatly affected by environmental and perceptual variations). Many variables only became apparent during the development of the project.

Furthermore, the project participants are mainly male because they are more affected by autism spectrum disorder. We have no data on the effectiveness of activities with girls. The sample turns out to be a portion of the target population of the survey that is analysed with the hope of obtaining information that can be extended to the entire population. In fact, even in the case of surveys carried out on a sample, the objective of a survey is always to improve knowledge of the target population (external validity).

Among the strengths of this work, there is certainly the breadth of the sample. Although it has room for improvement, thanks mainly to the work still being carried out, it is certainly much larger than those of the research studies examined. For example, Chambers and Radley, in their work on the training of football skills for adolescents with autism spectrum disorder through peer-mediated behavioural skills training, examined only three athletes [15]. The same can be said for Baulieu et al.'s [16] case study on social skills. Despite the fact that the research provided an excellent method and important docimological insights, it would be necessary for this work to repeat the research by enlarging the sample and thus supporting the results to a greater extent, even with a statistically relevant sample. In fact, the general rule is that the larger the sample size, the greater its statistical significance, i.e., the lower the probability that the results were obtained by pure coincidence. The works with the most sustained experimental samples are of a medical and biological-neuroscientific nature, thus more exploratory than educational [18–20,27]. In light of this, we can argue that the present data collection has not only original characteristics in its development but is also innovative given the large number of children on whom the intervention was carried out.

As stated, this study, which is still in progress, deserves further investigation to assess the maintenance of the autistic subjects' abilities in the long term. It is intended, moreover, to associate systematic qualitative observation that turns attention to the behaviour actually put "on the field" in a specific situation, faithfully recording the information sought, in order to correlate it with the tests administered.

## 10. Conclusions

In light of the results obtained, we can say that we have confirmed the original hypothesis. The principal results have shown that continuous and regular engagement in motor activities has positive health outcomes for children with autism in terms of reduced symptoms and improved quality of life. In conclusion, motor interventions have been shown to positively affect executive functions, which are closely related to the reduction of stereotypical behaviors. Therefore, enhancing attention and memory improves the behavioral area, with positive effects on children's quality of life as well. The results of the latest tests (Tables 6 and 7) show us that the typical difficulties of individuals with autism in shifting their attention between external stimuli and internal dialogue (due to poor communication between regions of the attentional network) can be improved through outdoor motor activities. The secondary results, on the other hand, show how it is possible to create social network systems, closely interconnected through a common goal: the well-being of children, in this specific case, with autism spectrum disorder. Local health authorities are collapsing, too often unable to handle the high number of requests for diagnosis first and rehabilitation intervention later. Activities such as this manage to offer a multi-component service aimed not only at children (educational intervention), families (supportive intervention), and local health authorities (supplementary—through sharing test results and periodically updating children's records).

On 28 July 2021, the then Minister of Labor and Social Policies, Andrea Orlando, spoke at the meeting of the Social Inclusion Network, pointing out that 'The Social Inclusion Network is the institutional forum for sharing social services with institutional actors, the Third Sector, and all the Social Partners'.

Moreover, in line with the 2030 Agenda, this study shows how a dialogue between territorial services can strengthen the social network in the provision of educational, sports, inclusive, and sustainable services but also positively intervene on the social mentality by developing inclusive acts [15].

In the Italian education and school system, unfortunately, to this day, despite its broad scientific scope, motor activity is viewed with reticence by teachers and often sacrificed in the name of frontal teaching. This work hopes to demonstrate how structured motor activities can support mnestic, attentional, and social processes in favor of learning.

**Author Contributions:** S.M.: Author of Section 2, Section 4, Section 5, Section 8, Section 9, and Section 10; F.P.C.: Author of the introduction and Section 3; F.T.: Author of Sections 6 and 7 and contributed to test administration; D.T.: coordinator. All authors have read and agreed to the published version of the manuscript.

**Funding:** The project was financed with funds from the Niccolò Cusano University.

**Institutional Review Board Statement:** The study was conducted in accordance with the Declaration of Helsinki, nd approved by the Institutional Review Board of Niccolò Cusano University in 1 September 2018.

**Informed Consent Statement:** Informed consent was obtained from all subjects involved in the study.

**Data Availability Statement:** Morsanuto S. https://doi.org/10.7346/-feis-XX-01-22_25; Cineca; researchgate.net; Peluso Cassese F.: https://orcid.org/0000-0002-9323-1005; Cineca; researchgate.net; Tafuri F. https://orcid.org/0000-0003-0784-1454; Tafuri D. https://ricerca.uniparthenope.it; Cineca.

**Conflicts of Interest:** The authors declare no conflict of interest.

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
