# Peer review of "Outdoor Education, Integrated Soccer Activities, and Learning in Children with Autism Spectrum Disorder: A Project Aimed at Achieving the Sustainable Development Goals of the 2030 Agenda"

_sustainability, doi:10.3390/su151813456_

Round 1
Reviewer 1 Report
Dear Authors,
your paper seems really praiseworthy since it describes a great project about the education of children with autism spectrum disorders. Moreover, it points out the importance of Agenda 2030 and it calls the scientific community to plan the achievement of its objectives.
The article is well structured and in line with the aim of this Journal. Some minor aspects have to be better addressed.
Title and abstract are clear and concise. With regard to the keywords, I suggest tu use some more keywords different from those used in the title for improving the visibility of this paper.
Introduction, methods and results are well presented. Just some minor corrections, as follows:
Line 84-85: please remove commas.
Line 171: the provided spaces instead of spaces provided
Regarding the discussion, I suggest to briefly integrate it with a final short paragraph about the possible positive lapels of your findings.
Conclusion and references are adequate.
Best regards and good luck
Author Response
Dear reviewer, thank you for the valuable suggestions. The revisions have all been made. Hoping we have met the requirements.
Thanking you
kind regards
Reviewer 2 Report
This paper entitled: “Outdoor Education, Integrated Soccer Activities, and Learning in Children with Autism Spectrum Disorder: a project aimed at achieving the goals in the sustainable development of Agenda 2030” is very interesting and provides important information to the scientific community. Moreover, subject of the research is exceedingly actual and important.
I recommend the authors minor revision in order to improve their manuscript:
- Abstract. The authors should start with a short intro that better highlights their work.
- Introduction is well written and clearly describe the scientific evidence that supports the hypothesis they have raised.
- Method. Authors should better define inclusion and exclusion criteria.
- Discussion is enriched with the existing theory. The authors well highlighted the scientific evidence that supports their findings. But, they should start with a first paragraph describing the main aims and then the main results.
- References: good number and up-to-date.
Author Response

(The authors gave the same response as above.)

Reviewer 3 Report
Please explain some issues,
Please confirm the subject of the study, there are confusing descriptions in the manuscript:
"The main objectives are to provide an educational and sports service in an area poorly provided with opportunities for children with intellectual disabilities to modify their predominantly sedentary condition and, through corporeality, to improve learning and autonomy processes. “ (lines 12-16)
”The project adheres to the concept of empowerment in sports for people with disabilities resting on awareness of one's own skills and perception of self-efficacy.“ (ines67-68)
”The study, led by Ditza Antebi-Zachor demonstrates the positive impact of outdoor physical activity in children with autism.“ (lines 85-86)
I feel the abstract is not closely related to the content of the main text.
Please make a targeted discussion on the subject (children of Autism) of this study based on the results.
No
Author Response
Dear reviewer, thank you for the valuable suggestions. The revisions have all been made. Hoping we have met the requirements.
Subject of study and objectives have been precisely outlined. The theoretical framework deepened. Efforts were made to better relate the abstract to the research content, as well as the discussion.
After the corrections were finished, the English language was further revised
Thanking you
kind regards
Reviewer 4 Report
Thank you very much for giving me the opportunity to review this manuscript. The idea of your article is interesting, my recommendations are the following:
Abstract
it would be recommended to add the city of “Frostine” (line 13). It´s unknown for most people.
It would be recommended the objectives of the agenda. Is needed people know it? If not, you should explain it.
It would be recommended to explain why play soccer is a good activity for this population. Is there any specific reason or is it a matter of having this option? Health and well-being do not appear.
What is “108 statistical units”(line 20)? It´s not clear
It´s needed to explain the sex of the participants. In fact, the agenda 2023 explain how necessary is to show the number of girls (number 5, equality).
Introduction
It would be recommended to explain why is important to promote soccer project. Is only to improve motor activity? The sport or physical activity effect is not only related to motor improvement. Thus, is this the only aim?
Is this research only about boys (lines 34, 36)? If there is not research with girls you should explain it. In fact, you should explain the differences between sex in this disorder, because it is important in this topic.
There are spelling and grammatical errors
If you add in hypothesis to improve cognitive skills you should explain it in the abstract and in the introduction. Consistency is not found.
Methods
It is recommended add specific sections (procedure, participants, tools…). The information is not clear.
If you put “children” sex is unknown. It´s necessary to add how much girls and boys conform the sample.
It is needed to include tool´s reliability, validity… It´s unknown
Results
The control group is unknow prior to this point. It´s necessary to specify the number from the abstract and the sexes of the participants.
The “figure 2 or 3” is a table. This contains raw data that could be worked on in depth. The tables should not be as is the data output but must be worked to explain the results.
Discussion
It would be recommended to expand this section quite a bit. There is not a wide contrast of the theme.
Conclusion
It would be advisable to clarify and delimit conclusions.
References
It would be recommended to extend the bibliographic search. There are relevant jobs not reflected.
Author Response
Dear referee thank you for the valuable suggestions. The changes have all been made. I hope they meet the requirements.
Subject of study and objectives have been precisely outlined. The theoretical framework deepened.
The project design has been better defined. The sample accurately described (Control group, experimental, gender in accordance with Art. 5 of the 2023 Agenda)
Efforts were made to better relate the abstract to the research content, as well as discussions and conclusion.
The tables were not raw data, but already processed (with SPSS), they were deepened and explained better
After the corrections were finished, the English language was further revised.
Thanking you
Best regards
Round 2
Reviewer 3 Report
No
NO
Author Response
Dear Sir, as requested, I have implemented the introduction, research design, discussion and conclusions. I hope the changes I have made are satisfactory.
Kind regards
Stefania Morsanuto